# Impact of gravity wave drag on the thermospheric circulation: Implementation of a nonlinear gravity wave parameterization in a whole atmosphere model

Yasunobu Miyoshi [1], Erdal Yiğit [2]

[1] Department of Earth and Planetary Sciences, Kyushu University, Fukuoka, Japan
[2] Department of Physics and Astronomy, Space Weather Lab, George Mason University, Fairfax, VA, USA

*Correspondence to*: Yasunobu Miyoshi (y.miyoshi.527@m.kyushu-u.ac.jp)

**Abstract.** To investigate the effects of the gravity wave (GW) drag on the general circulation in the thermosphere, a nonlinear GW parameterization that estimates the GW drag in the whole atmosphere system is implemented in a whole

atmosphere general circulation model (GCM). Comparing the simulation results obtained with the whole atmosphere scheme with the ones obtained with a conventional linear scheme, we study the GW effects on the thermospheric dynamics for solstice conditions. The GW drag significantly decelerates the mean zonal wind in the thermosphere. The GWs attenuate the migrating semidiurnal solar tide (SW2) amplitude in the lower thermosphere, and modifies the latitudinal structure of the SW2 above 150 km height. The SW2 simulated by the GCM based on the nonlinear whole atmosphere scheme agrees well

with the observed SW2. The GW drag in the lower thermosphere has zonal wavenumber 2 and semidiurnal variation, while the GW drag above 150 km height is enhanced in high latitude. The GW drag in the thermosphere is a significant dynamical factor and plays an important role in the momentum budget of the thermosphere. Therefore, a GW parameterization accounting for thermospheric processes is essential for coarse-grid whole atmosphere GCMs in order to more realistically simulate the atmosphere-ionosphere system.

## 1 Introduction

It has been widely recognized that internal atmospheric waves from the lower atmosphere, such as planetary waves, solar tides and gravity waves (GWs), propagate into the upper atmosphere and affect the circulation in the thermosphere-ionosphere system (Yiğit and Medvedev, 2015, and references therein). In this study, we focus our attention on the impact of GWs of lower atmospheric origin on the thermospheric circulation and solar tides. While middle atmospheric effects of GWs

have been extensively studied, GW effects in the upper atmosphere above the turbopause has been studied to a much lesser extent due to a combination of observational and modelling challenges. On one hand, due to insufficient observations of the neutral winds in the thermosphere, the behaviour of GWs in the thermosphere has not been sufficiently known. On the other hand, whole atmosphere models, which can study GW propagation continuously in different layers of the atmosphere, has been developed only in recent times. Middle atmosphere models had upper boundaries somewhere in the upper mesosphere,

while thermosphere-ionosphere models had lower boundaries around the lower thermosphere. Additionally, the vast majority of the existing GWs focused on the middle atmosphere dynamics and thus were not designed to represent GW processes in the upper atmosphere. Developments and challenges in the parameterization of gravity waves in the whole atmosphere region has been discussed in detail in the work by Yiğit and Medvedev (2013). Briefly, gravity waves propagating from the lower atmosphere into the thermosphere are subject to additional dissipation processes that are characteristic of the thermosphere-ionosphere system as documented for the first time in the context of a parameterization in the work by Yiğit et al. (2008). These wave damping mechanisms are molecular diffusion and thermal conduction and ion drag in additional to nonlinear interactions. Therefore, in order to realistically simulate GW dynamics in the thermosphere, these processes have to be taken into account. The majority of the conventional GW schemes were not designed for the upper atmosphere in the first place.

Recently, an increasing number of numerical studies have revealed direct upward propagation of GWs from the lower atmosphere into the thermosphere and demonstrated significant GW effects on the thermospheric circulation (e.g., Yiğit et al., 2014; Heale et al., 2014; Gavrilov and Kshevetskii, 2015). Earlier, using a regional model and ray-tracing method, Vadas and Fritts (2004) showed that GWs generated by cumulus convection can propagate into the thermosphere and produce large GW drag in the thermosphere. GWs with high frequency (large vertical wavelength) can penetrate into the thermosphere (Vadas and Fritts, 2005). Yiğit et al. (2008) developed a nonlinear whole atmosphere GW parameterization, and succeeded in the implementation and application of their GW parameterization in the Coupled Middle Atmosphere Thermosphere-2 (CMAT2) general circulation model (GCM). Yiğit et al. (2009) showed that the dynamical effects of gravity waves in the thermosphere are comparable with the ion drag effects up to ionospheric F2 region altitudes. Later, based on the similar modeling framework, Yiğit and Medvedev (2009) showed for the first time that GW thermal effects are very important globally in the thermosphere, competing with Joule heating, and ultimately cool the thermosphere. More recently, Yiğit and Medvedev (2017) demonstrated that the small-scale GWs impact the amplitude of the diurnal tide in the low-latitude middle atmosphere and in the high-latitude thermosphere. Using the first generation CMAT model with two different older GW parameterizations, England et al. (2006) investigated effects of the GW drag on the diurnal tide and green line airglow emissions during equinox in the equatorial mesosphere and lower thermosphere (MLT). Based on idealized numerical simulations with the Yiğit et al. (2008) scheme, Medvedev et al., (2017) have discovered that the magnetic field configuration can significantly influence the propagation and dissipation of lower atmospheric GWs in the thermosphere via the ion drag force.

GW effects can also be studied using high-resolution GCMs. Models are increasingly capable of implementing higher resolutions, which can capture smaller scale physics. Using a GW-resolving (i.e., high horizontal resolution) GCM, Miyoshi and Fujiwara (2008) and Miyoshi et al. (2014, 2015) investigated upward propagation of GWs and the GW drag in the thermosphere. They indicated that the GW drag in the thermosphere is much larger than that in the mesopause region. The GW activity in the thermosphere is stronger in winter than in summer and is correlated with the strength of the strato-mesospheric jet. Using a GW-resolving WACCM, Liu et al. (2014) also studied upward propagation of GWs excited by

tropical convection up to 105 km height. Overall, high-resolution simulations supported the finding that the mean GW effects in the thermosphere can be adequately represented by physics-based GW parameterizations, such as the one developed in the work by Yiğit et al. (2008).

Previous numerical studies indicated that the GW drag plays an important role in maintaining the momentum and energy balance in the thermosphere. Both GW parameterizations and high-resolution simulations provide various advantages as well as some limitations. While, the mean global structure of GW effects is well represented by GW parameterizations extending into the thermosphere, high-resolution simulations can more self-consistently simulate GW processes probably in more detail, for example, smaller-scale variability in GWs can be better captured. This implies overall that a GW resolving GCM is necessary in order to simulate themopheric circulation more accurately. However, numerical diffusion schemes (e.g., hyperdiffusivity) may excessively damp smaller scale GWs. Often, GW sources and their generations are still parameterized in high-resolution simulations. Also, conducting numerical simulations with a GW-resolving GCM requires high performance computer systems and overall needs much more computational time and data storage. Therefore, long-term simulations using a GW-resolving GCM is unpractical. Therefore, a low-resolution GCM based on a physics-based whole atmosphere GW parameterization is strongly required. However, there are only a few studies concerning GW drag parameterization for the thermosphere and there are various aspects of GW effects in the thermosphere that are still unexplored. One such unexplored territory that is the focus of this paper is the interaction of GWs with the semidiurnal migrating tide (SW2) in the upper atmosphere. Simultaneously, this work serves as the first study with the Whole Atmosphere GCM by Miyoshi and Fujiwara (2003) implementing the whole atmosphere GW parameterization by Yiğit et al. (2008). So, we will also study and revisit the mean GW effects on the thermosphere and solar tides.

The descriptions of the GCM, the GW parameterization, and of numerical simulations are presented in section 2. Results and Discussion are presented in section 3. Concluding remarks follow in section 4.

## 2 General Circulation Model, Gravity Wave Schemes, and Experiment Design

The model used in this study is a whole atmosphere GCM as shown in Miyoshi and Fujiwara (2003, 2006, 2008). This model is a thermospheric extension of the middle atmosphere model developed at Kyushu University (Miyahara et al., 1993; Miyoshi, 1999). The GCM is a global spectral model with a horizontal grid spacing of $2.8°$ latitude $\times 2.8°$ longitude. The GCM has 150 layers with a vertical resolution of 0.2 scale heights. The GCM covers the region from the ground to the exobase. It has a complete set of physical processes appropriate for the whole atmosphere region. The GCM is the same as the neutral atmospheric part of an atmosphere-ionosphere coupled model GAIA (Ground-to-topside model of Atmosphere and Ionosphere for Aeronomy; Jin et al., 2008, 2011).

The GCM incorporates schemes for a hydrological cycle, a boundary layer, moist convection, and infrared and solar radiations (Miyoshi and Fujiwara, 2003, 2006). Effects of mountains and land-sea-contrast are also taken into account. The GCM was nudged by Japanese Meteorological reanalysis data (JRA55; Kobayashi et al., 2015) up to 40 km height to

simulate realistic temporal variations in the lower atmosphere (Jin et al., 2011). In the thermosphere, the GCM has schemes for molecular diffusion, thermal conductivity, Joule heating, ion-drag force, and auroral precipitation heating. To estimate Joule heating, ion-drag force, and auroral precipitation heating, the electron density are prescribed using an empirical ionosphere model. The global electron density distribution produced by the solar radiation is represented by the Chiu's empirical model (Chiu, 1975). Electrons produced by auroral particles are estimated by Fuller-Rowell and Evans (1987). We use a coarse grid in this study, which provides computational efficiency, and represent GWs that are not explicitly resolved by the model with orographic and nonorographic GW parameterizations. The GW parameterization developed by McFarlare (1987) is used for orographic GWs. The previous standard version of the GCM includes a linear nonorographic GW parameterization developed by Lindzen (1981). However, the GW drag estimated by these GW parameterizations are taken into account only below 100 km height. Thus, note that no GW effects are calculated in the thermosphere above 100 km height in this configuration. This setup mimics the traditional approach of accounting for GWs only in the middle atmosphere, which is essentially what low-top middle atmosphere models used to do. The numerical simulation using this original GCM is called EXP1. This standard version is described in detail in previous publications (Miyoshi and Fujiwara, 2003, 2006; Miyoshi et al., 2009).

To assess impacts of GW drag on the general circulation in the thermosphere as well as in the lower and middle atmosphere, we need a GW scheme that extends into the thermosphere. Therefore, the GW parameterization developed in the work by Yiğit et al., (2008) has been implemented in the GCM developed by Miyoshi and Fujiwara (2003). Yiğit's GW parameterization can estimate the GW effects in the whole atmosphere system from the troposphere to the upper thermosphere. The GW spectrum is specified in terms of momentum fluxes as a function of horizontal phase speeds. The phase speeds of GWs used in GW calculations range from 2 m s$^{-1}$ to 80 m s$^{-1}$. The peak GW flux at the source level and the horizontal wavenumber are set at 0.00025 m$^2$ s$^{-2}$ and $2\pi/250$ km$^{-1}$, respectively (see Figure 1 of Yiğit et al., 2012 for a representative GW spectrum). The GW spectrum adopted in this study and its relation to the observation were discussed in detail in Yiğit et al. (2008). Dissipation of gravity waves due to nonlinear interactions (Medvedev and Klaassen, 2000), radiative damping, molecular diffusion and thermal conduction, eddy viscosity and ion drag are taken into account. This scheme has been used successfully in different Earth modeling frameworks (e.g., Lübken et al, 2018) and also for Mars atmosphere (Yiğit et al., 2018). The numerical simulation with Yiğit's whole atmosphere parameterization is called EXP2. The GCM used in EXP1 is identical to the GCM used in EXP2 except for the nonorographic GW parameterization. Namely, in EXP2, Lindzen's GW parameterization, which cuts off GW effects at around 100 km, is replaced by Yiğit's GW parameterization, which calculates GW effects in the entire atmosphere. By comparing EXP1 and EXP2, we investigate the impact of the GW drag on the general circulation in the middle and upper atmospheres by comparing EXP1 and EXP2.

To exclude influences from temporal variations in solar UV/EUV fluxes and geomagnetic activity, we performed the numerical simulations under solar minimum and geomagnetically quiet conditions. The 10.7 cm solar radio flux (F10.7) was fixed at $70 \times 10^{-22}$ W m$^{-2}$ Hz$^{-1}$, and the cross polar potential was set at 30 kV during the numerical simulations. Numerical simulations was conducted under June solstice conditions. Numerical simulation started on 1 June (Year 2015), and a 2-year

numerical integration with seasonal variation was performed. The time step of the GCM is 30 s, and the GW drag is estimated at each time step. The data are sampled every 1 h during the numerical simulation. The data from 1 June to 30 June in the second year (Year 2016) are analysed in this study.

## 3 Results and Discussions

### 3.1 Zonal mean fields

Impacts of GWs on the zonal mean zonal wind are examined first. Figure 1a shows the height–latitude section of the zonal and diurnal mean zonal wind obtained by the application of the Lindzen scheme (EXP1). Data are averaged from 1 June to 30 June. Note that thermospheric GW effects above 100 km height are not incorporated in this scheme. Strong jets exist in the stratosphere and mesosphere. These jets weaken in the upper mesosphere, and the reversal of the zonal wind direction
occurs around 80−100 km height. It is well known that this reversal of the zonal wind is generated by the GW drag (e.g., Lindzen 1981, Matsuno 1982, Garcia and Solomon, 1985). Again, the westward and eastward wind appear above 120 km height in the Northern Hemisphere (NH) and Southern Hemisphere (SH), respectively. The peak of the westward wind (48–52 m s$^{-1}$) above 120 km is located at 50° N, whereas the peak of the eastward wind (20–25 m s$^{-1}$) above 120 km height appears at 30° S.

Figure 1b shows the zonal and diurnal mean zonal wind distribution obtained by the application of the Yiğit scheme (EXP2). As shown before, Yiğit's GW parameterization is implemented in the whole atmosphere region. The strato-mesospheric jets weaken in the upper mesosphere, and the reversal of the zonal wind direction occurs at 80−100 km height. The difference of the zonal mean zonal wind between EXP2 and EXP1 is shown in Figure 1c. There are substantial differences of the magnitudes of the strato-mesospheric jets between EXP1 and EXP2. The eastward jet in the SH is stronger in EXP1 than in
EXP2. On the other hand, the westward jet in the NH at 20–50° N is stronger in EXP1 than in EXP2 by ∼22 m/s, and the westward jet poleward of 50°N is weaker in EXP1 than in EXP2 by ∼8 m/s. These differences of the strength of the jets are mainly caused by the differences of the GW drag distributions as shown later (section 3.4). Above 120 km height, the westward and eastward winds are dominant in the NH and SH, respectively. These features obtained in EXP2 are the same as those in EXP1. However, the reversal of the zonal wind direction at 80−100 km height is much clearer in EXP2 than in
EXP1. The peak values of the zonal mean zonal wind above 120 km height are weaker in EXP2 than in EXP1. These results indicate that including GW effects above 100 km height affects the magnitude of the zonal mean zonal wind and provides a deceleration mechanism.

Figures 2a and 2b show the height–latitude distribution of the zonal mean meridional wind obtained by EXP1 and EXP2, respectively. In both experiments, southward flow from the summer pole to winter pole is dominant at 50–100 km height,
whereas northward flow appears between 100 and 120 km height. These flows are stronger in EXP2 than in EXP1, which is explained by the enhanced GW drag in EXP2 as shown later. Above 130 km height, southward flow is dominant in both

experiments. The magnitude of the southward wind between 130 and 250 km height is weaker in EXP2 than that in EXP1 except for southward of 30° S (Figure 3c). This weaker meridional wind in EXP2 is caused by the meridional component of the GW drag. On the other hand, the difference of meridional wind between EXP1 and EXP2 is small above 250 km height (less than 10 %).

Figures 3a and 3b shows the height-latitude distribution of the zonal mean temperature obtained by EXP1 and EXP2, respectively. At 80–100 km height, Cooling and warming occur at 30–90° N and at 60–90° S, respectively (Figure 3c). This cooling and warming is caused by the enhanced southward wind (meridional circulation) at 80–100km height in EXP2. Namely, the cooling (warming) at 30–90° N (60–90° S) is due to enhanced upward (downward) wind. It is noteworthy that cooling prevails above 100 km height. In particular, cooling at high latitudes in the NH exceeds 60 K. This cooling is caused

by the GW thermal effect. This indicates that GW induced cooling also affects thermal structure in the upper thermosphere. Our results support the conclusion of the GCM work by Yiğit and Medvedev (2012), who showed for the first time that GWs cool the thermosphere during low solar activity conditions.

## 3.2 Migrating Semidiurnal Tide

The impact on the migrating semidiurnal tide (SW2) is examined here. Figure 4a shows the height–latitude distribution of the temperature component of the SW2 amplitude in June obtained by EXP1. The amplitude maximizes around 125 km height. The maxima are 41 K at 15° S and 38 K at 20° N. The peak of the SW2 amplitude above 200 km height (34 K) appears at 15–25° S, and the secondary peak (10 K) is found around 45° N. Using the SABER (Sounding of the Atmosphere using Broadband Emission Radiometry) measurement on board the TIMED (Thermosphere Ionosphere Mesosphere

Energetics Dynamics) satellite, Pancheva (2011) studied climatology (6-year mean from 2002 to 2007) of SW2 temperature tide observation. Figure 2.3 in Pancheva (2011) indicates that the SW2 in June at 110 km height has peaks at 15–30° N and 15–25° S. The maxima at 15–30° N and at 15–25° S are 25–28 K and 15–20 K, respectively. The peak values at 110 km height in EXP1 are about 30–35 K, which is much larger than the observation. Forbes et al. (2011) investigated seasonal variation of the SW2 in the exobase (about 400–500 km) using the CHAMP (Challenging Minisatellite Payload) and

GRACE (Gravity Recovery and Climate Experiment) accelerometer measurements. The observed SW2 in June also has two peaks (15–20° S and 40° N), and the peak value during solar minimum is 24 K (Figures 7 and 8 of Forbes et al., 2011). The simulated SW2 in the SH (NH) is larger (smaller) than the observed SW2.

Figure 4b shows the temperature component of the SW2 amplitude obtained by EXP2. The SW2 in the lower thermosphere maximizes at 15–20° S and at 20° N. The peak values of the monthly mean SW2 amplitude in EXP2 at 110 km height are 26

K at 20° N and 21 K at 15° S. The SW2 peaks at 100–200 km height moves southward in EXP2. The SW2 amplitude in the lower thermosphere in EXP2 is weaker than the one in EXP1 by about 20–40 % (Figure 4c). Results obtained with EXP2 compares better with the observed SW2 amplitude. Above 200 km height, the SW2 amplitude in EXP2 maximizes at 10° S

(25 K), and secondary peak is found at 40° N (15–20 K). The SW2 amplitude in the SH (NH) is weaker (stronger) in EXP2 than in EXP1. This means that the latitudinal structure of the SW2 above 200 km height is modified by GW propagating and dissipation in the middle and upper thermosphere. Moreover, the SW2 in the upper thermosphere obtained by EXP2 agrees well with the observed SW2 amplitude.

Figures 5a and 5b show the zonal wind component of the SW2 amplitude in EXP1 and EXP2, respectively. Figure 5c shows the amplitude difference between the EXP1 and EXP2. The maximum of the zonal wind component of the SW2 at 120 km height in EXP1 (EXP2) is 68 (55) m s$^{-1}$. The GW parameterization attenuates the zonal wind component of the SW2 in the lower thermosphere. Moreover, above 200 km height, the GW parameterization modifies the latitudinal structure of the zonal wind component. The effects of the GW drag on the migrating diurnal tide in the MLT was studied by Miyahara and

Forbes (1991). They showed that the DW1 is attenuated by the GW drag. Yiğit and Medvedev (2017) investigated the effects of GWs on the diurnal tide from the mesosphere to the upper thermosphere. They found that while GWs enhance the tidal amplitude in the MLT, GWs can both damp and strengthen the tides in the thermosphere. The impact on DW1 obtained in this study is similar to that in Yiğit and Medvedev. On the other hand, the present results indicate the GW drag has widereaching implications for the migrating tides. Namely GWs attenuate the SW2 amplitude in the MLT, improving model

simulations with respect to observations. Overall, this is the most dominant effect of the GW drag on the SW2.

The SW2 also has significant day-to-day variations. For example, the SW2 amplitude at 20° N in EXP1 ranges from 27 to 37 K, whereas the SW2 amplitude in EXP2 ranges from 22 to 31 K. The standard deviation of day-to-day variations in the SW2 amplitude at 20° N in EXP1 and EXP2 are 2.8 K and 2.9 K, respectively. Similar day-to-day variations in the SW2 amplitude are found below 100 km height. These results indicate that day-to-day variations in the SW2 amplitude are

primarily generated in the lower atmosphere and propagate into the lower thermosphere.

### 3.3 Migrating terdiurnal tide

Figure 6a shows the height–latitude distribution of the temperature component of the migrating terdiurnal tide (TW3) amplitude in June obtained by EXP1. The amplitude peak is located at 15° N latitude, and secondary peak appears at 25–30° S. The maxima are 23 K at 15° N and 130 km height, and 18 K at 17.5° S and 165 km height.

Figure 6b shows the temperature component of the TW3 amplitude obtained by EXP2, and Figure 6c shows the amplitude difference between EXP1 and EXP2. The latitudinal structure of the TW3 in EXP2 is quite similar to that in EXP1. However, the amplitude is weaker in EXP2 than in EXP1 by about 20–40%. The amplitude difference is significant in the 120–220 km height range. The TW3 is also attenuated by the GWD. Forbes et al. (2008) indicated that the TW3 amplitude at 110 km height is between 5 and 8 K. However, there are few studies

concerning the satellite observation of TW3 amplitude in the 120–220 km height range. A detailed comparison of the TW3 amplitude between the simulation and observation is a subject of a future study.

### 3.4 Zonal mean of the zonal GW drag

Figure 7a shows the height–latitude section of the zonal and diurnal mean of the zonal GW drag estimated by Lindzen's parameterization (EXP1). Eastward (westward) acceleration exists in the NH (SH), and attenuates the mesospheric jet. Figure 7b shows the zonal and diurnal mean of the zonal GW drag estimated by Yiğit's parameterization (EXP2). The differences of the GW drag below 100 km height are substantial. The magnitude of the GW drag in EXP1 below 100km height is similar to that in EXP2. However, the peak of the GW drag in EXP1 is located around 60–70 km, whereas that in EXP2 is located around 90–100 km height. These differences of the GW drag below 100 km height produce the differences of the strato-mesospheric jets. The GW drag in EXP2 extends to 300 km height. It is noteworthy that the magnitude of the GW drag in 150–300 km height is comparable to that in the MLT.

### 3.5 Longitudinal variation of the GW drag at 35° N

In the previous sections, we investigate zonal and diurnal mean of the GW drag. Longitudinal and diurnal variabilities of the winds are significant in the thermosphere, so that longitudinal/diurnal variability of the GW drag is examined next. Figure 8a shows height-longitude distribution of the zonal GW drag at 35° N, where the SW2 amplitude maximizes in the lower thermosphere. The zonal GW drag in Figure 8a is averaged between 00 UT and 01 UT. The GW drag at 70–100 km height is eastward at all longitudes, and contribute the attenuation and reversal of the westward jet in the upper mesosphere.

The zonal and diurnal mean of the zonal GW drag at 35° N in the 100–200 km height region is smaller than 20 m s$^{-1}$ (day)$^{-1}$ (Figure 7b). However, Figure 8a indicates that the GW drag can range from $-100$ m s$^{-1}$ (day)$^{-1}$ to 200 m s$^{-1}$ (day)$^{-1}$ within the one-day period. The maximum acceleration is located at 157° E and 150 km height. The GW drag has zonal wavenumber 2 structure in the 100–200 km height region, and the peak of the GW drag descends with increasing longitude. Figure 8b shows a height-longitude section of the zonal wind at 35 N° averaged between 00 UT and 01 UT. The zonal wind distribution in the 100–200 km height region has also zonal wavenumber 2 structure, and descends with increasing longitude, indicating characteristics of the upward propagating SW2. The eastward (westward) acceleration of GW drag in the 100–200 km region occurs in the region of westward (eastward) wind. It is clearly seen that the GW drag attenuates the zonal wind variation associated with the SW2. Thus, the attenuation of the SW2 in EXP2 is explained by dissipating GWs as represented by the Yiğit scheme. The main dissipation mechanisms of GWs in the thermosphere are due to ion drag, molecular diffusion and thermal conduction, while in the MLT nonlinear interactions play an important role.

Figure 9a and 9b show the global distribution of the zonal GW drag at 120 km at two representative times, 00–01 UT and at 06–07 UT, respectively. The GW drag in low and middle latitudes has zonal wavenumber 2 structure. The magnitude of the GW drag sometimes exceeds 150 m s$^{-1}$ (day)$^{-1}$. The distribution of GW drag in 00–01 UT is clearly out of phase with that in 06–07 UT, indicating semidiurnal variation of the GW drag. In the work by Miyoshi and Fujiwara (2014), the relationship between GW drag and SW2 was investigated using a GW-resolving GCM. They showed that the semidiurnal variation of the

GW drag is significant in the lower thermosphere and it decelerates the background zonal wind variation. The present result is consistent with the result obtained by Miyoshi and Fujiwara (2014).

### 3.6 Longitudinal Variation of the GW Drag at High Latitudes

In this section, the relationship between the GW drag and the zonal wind in high latitudes, where the diurnal variation of the zonal wind is the largest, is investigated. Figure 10a shows height–longitude section of the zonal GW drag in EXP2 at $65°$ N in 00–01 UT. Eastward acceleration is dominant in the 60–110 km height region, which attenuates the westward wind. Above 150 km height, zonal wavenumber 1 structure is dominant. Eastward acceleration (westward acceleration) appears in 0–$180°$ E (180–$360°$ E) longitude sector. Studying the GW drag together with the zonal wind (EXP2) distribution shown in Figure 10b shows that the GW drag is predominantly directed against the zonal wind and thus tends to decelerate the wind. It is noteworthy that the magnitude of eastward acceleration at 130–270 km height is a few hundred m s$^{-1}$ (day)$^{-1}$, and the maximum values of $-650$ m s$^{-1}$ (day)$^{-1}$ is found at $275°$ E and 230 km height.

To investigate the impact of the GW drag on the zonal wind variation, the zonal wind distribution obtained by EXP1 is shown in Figure 10c. In EXP1, the westward (eastward) wind prevails in 0–$180°$ (180–$360°$) longitude sector above 150 km height. Figure 10d shows the difference of the zonal wind between EXP2 and EXP1 (EXP2–EXP1). This essentially shows the difference between the impact of the GW effects on the zonal circulation represented by two different schemes. It is noteworthy that the differences are substantial not only above 100 km, where EXP1 does not include any GW drag, but also in the mesosphere. Both the westward and eastward wind above 150 km height are 10–30 m s$^{-1}$ smaller in EXP2 than in EXP1. This means that the GW drag attenuates the amplitude of the wave number 1 structure of the zonal wind. The difference of the zonal wind in the mesosphere is mainly caused by the substantial differences of the treatment of the GW process. This differences will be discussed in section 3.7.

The global distribution of the zonal GW drag at 200 km height is examined here. Figure 11a and 11b show the zonal GW drag distribution in 00–01 UT and 06–07 UT, respectively. In both figures, the GW drag is significant at high latitudes. For example, in 00–01 UT, westward acceleration of 1500 m s$^{-1}$ (day)$^{-1}$ is found at 40–$50°$ E and $80°$ N, while eastward acceleration of $-1200$ m s$^{-1}$ (day)$^{-1}$ appears at $275°$ E and $70°$ N. These strong GW drag regions move westward with time, and westward acceleration of $-1200$ m s$^{-1}$ (day)$^{-1}$ appears at $170°$ E and $80°$ N in 06–07 UT. The magnitude of the GW drag in high latitudes is comparable to the magnitude of the ion-drag force (e.g., Yiğit et al., 2012; Miyoshi et al, 2014).

Figure 11c and 11d show the horizontal wind distribution at 200 km height in 00–01 UT and 06–07 UT, respectively. Color shading in Figures 11a and 11b shows the zonal wind distribution. The enhanced zonal GW drag is located at the regions where the strong zonal wind appears. The strong zonal winds in high latitudes are mainly generated by the convective electric fields of magnetospheric origin, auroral energy precipitation, and ion-neutral coupling processes such as ion drag force and Joule heating (e.g., Yiğit and Ridley, 2011). The enhanced eastward (westward) wind is favorable for upward

propagation of westward (eastward) moving GWs from the lower atmosphere. The westward (eastward) acceleration due to the dissipation/breaking of westward moving (eastward) GWs occurs in the eastward (westward) wind region. This is the reason why the GW drag is enhanced at high latitudes. These results indicated that the GW drag plays an important role on the momentum budget in high latitudes around 200 km height. Using a GW-resolving GCM, Miyoshi and Fujiwara (2014) showed the enhancement of the GW drag in polar region at 200 km height. Their result is in good agreement with the results presented here.

## 3.7 Discussions

General circulation models (GCMs) provide a powerful methodology for studying the global effects of gravity waves (GWs) in the atmosphere. One strength is the continuous coverage of atmospheric layers, thus interaction processes between different layers can be studied. However, they have limited resolutions so physical parameterizations are crucial. Nowadays, atmospheric models are gradually being converted into whole atmosphere models, which can provide a framework in which atmospheric wave propagation can be studied from the lower atmosphere to the upper atmosphere in a more self-consistent manner (e.g., Miyoshi and Fujiwara, 2003, 2008; Akmaev et al., 2008). Also, it is increasingly acknowledged that GW parameterizations must cover the entire atmosphere, following the realization that GWs deposit their energy and momentum at different layers in the atmosphere with a significant portion being deposited in the middle thermosphere. In this context, we exploit the capability of the whole atmosphere GW parameterization of Yiğit et al. (2008). Note that recent studies showed that lower atmospheric GWs can directly propagate into the thermosphere and can dump significant energy and momentum there. For example, the magnitude of the GW drag in the lower thermosphere sometimes exceeds 150 m s$^{-1}$ (day) $^{-1}$, whereas the magnitude of the GW drag at 200 km height at high latitudes exceeds 1000 m s$^{-1}$ (day) $^{-1}$. In summary, GWs can produce a variety of effects in the thermosphere, including dynamical (Yiğit et al., 2009), thermal (Yiğit and Medvedev, 2009; Hickey et al., 2011), and mixing effects (Walterscheid and Hickey, 2012). Transient atmospheric processes can dramatically modulate penetration of GWs into the thermosphere: During minor warming, thermospheric GWs activity can be enhanced significantly (e.g., Yiğit et al., 2014; Yiğit and Medvedev, 2016), while during major warmings it may encounter a decrease (Nayak and Yiğit, 2019). Here, we used a whole atmosphere GCM incorporating a whole atmosphere GW parameterization in order to the study the mean dynamical effects of GWs and their impact on the semidiurnal tides in the thermosphere.

Using the SABER (Sounding of the Atmosphere using Broadband Emission Radiometry) measurement on board the TIMED (Thermosphere Ionosphere Mesosphere Energetics Dynamics) satellite, Pancheva (2011) investigated behavior of the SW2 in the MLT. There, the typical observed peak values for the SW2 amplitudes are situated at low-latitudes during June solstice: 25-28 K at 15–30° N and at 15–20 K at 15-25° S. In our study, GCM simulations based on two different GW parameterizations yield different results for the SW2 tide. While the simulation with the standard Lindzen scheme overestimate the tidal amplitude, the one with the Yiğit scheme matches observations better. This can be explained by the

additional GW drag in the thermosphere (i.e., additional physics) as accounted for by the Yiğit scheme, which attenuates the zonal wind variation associated with the SW2.

Using the CHAMP (Challenging Minisatellite Payload) and GRACE (Gravity Recovery and Climate Experiment) accelerometer measurements, Forbes et al. (2011) showed the SW2 amplitude in the upper thermosphere. The GCM without GW drag parameterization in the thermosphere fails to reproduce the observed SW2 in the upper thermosphere, whereas the GCM with the GW parameterization succeeds in reproducing the behaviour of the observed SW2 in the upper thermosphere. This result also indicates the importance of the GW effects in the thermosphere.

There are substantial differences between the linear and the nonlinear schemes in the treatment of GW processes as has been initially discussed in detail in the work by Yiğit et al (2008). One major difference is that the linear scheme is based on the linear saturation principle, ignoring wave-wave interactions, while the Yiğit scheme takes into account not only nonlinear wave-wave interactions but also dissipation of GWs due to additional processes, such as ion drag, molecular viscosity and thermal conduction are taken into account, which are important dissipative processes in the thermosphere-ionosphere system. Any GCM that is extending into the thermosphere, including a GW parameterization, must incorporate these effects on GW propagation. While the linear scheme assumes an artificial tuning factor for the GW drag, the nonlinear scheme does not require any artificial tuning parameters. However, GW parameterizations are not devoid of limitations. They all assume single-column approach and instantaneous response of the flow field to the upward propagating waves.

The whole atmosphere GCM uses an empirical ionospheric model. At high-latitudes, the behavior of the ionosphere can substantially influence the thermospheric circulation. On the other hand, the background atmosphere is very important for the GW propagation and dissipation. A modelling framework with self-consistent two-way coupled ionosphere-thermosphere system could provide a more realistic picture of ion-neutral coupling and GW effects could be evaluated more precisely at high-latitudes.

## 4. Concluding remarks

The GW parameterization developed by Yiğit et al. (2008) has been implemented in the Japanese Kyushu University whole atmosphere GCM (Miyoshi and Fujiwara, 2008), and the impact of small-scale GWs on the migrating semidiurnal tide as well as the GW effects on the general circulation of the thermosphere has been studied. We obtained the following results.

1. The GW drag attenuates the magnitude of the zonal/diurnal mean zonal wind in the thermosphere. The GW drag modifies the zonal mean meridional and temperature distributions in the thermosphere.

2. The GW drag attenuates the SW2 amplitude in the lower thermosphere, and modifies the latitudinal structure of the SW2 above 150 km height. The GW drag also attenuates the TW3 amplitude in the thermosphere.

3. The GW drag in the lower thermosphere has zonal wavenumber 2 structure and has semidiurnal variation.

4. The GW drag above 150 km height is enhanced in high latitudes. The maximum value sometimes exceeds 1000 m s$^{-1}$ (day) $^{-1}$. This means that the GW drag plays an important role on the momentum balance in high latitudes above 150 km height.

The whole atmosphere GCM used in this study uses an empirical ionosphere. Therefore, impacts of the GW drag on the ionospheric variability have not been investigated in this study. In the next step, implementation of GW drag parameterization in an atmosphere-ionosphere coupled model, such as GAIA (Jin et al., 2011) is strongly required. Using an atmosphere-ionosphere coupled model with GW drag, we will investigate impacts of the GW drag on the ionospheric variability.

**Data Availability.** Upon request, the data used for the publication of this study are available from Yasunobu Miyoshi (y.miyoshi.527@m.kyushu-u.ac.jp)

**Author Contribution.** YM performed the simulation and wrote a substantial portion of the paper. EY provided the whole atmosphere GW parameterization scheme, and significantly contributed to writing and discussion of results.

**Competing interests.** EY is one of the editors of this special issue. The authors declare that there are no conflicts of interests.

**Acknowledgement.** This work was supported by a JSPS Grant-in-Aid for Scientific Research grand numbers (B 15H03733) and (18H04447). Y. M. was partially supported by JSPS and DFG under the Joint Research Projects-LEAD with DFG (JRPs-LEAD with DFG). The GFD/DENNOU library was used to produce the figures. The numerical simulation was performed using the computer system at Research Institute for Information Technology of Kyushu University, and at National Institute of Information and Communication Technology, Japan. E. Y. was partially funded by the National Science Foundation (NSF) grant AGS 1452137.

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

**Figure Captions**

**Figure1:** (a) Height-latitude section of zonal and diurnal mean zonal wind obtained by EXP1 (the application of the Lindzen scheme below 100 km height). Data are averaged from 1 June to 30 June. Contour intervals of black lines are 10 m s$^{-1}$. Negative and positive values are eastward and westward winds, respectively. (b) As in Figure 1a except for the application of the Yiğit scheme in the whole atmosphere (EXP2). (c) Difference of the zonal wind between EXP1 and EXP2 (EXP2–EXP1). Contour intervals are 5 m/s.

**Figure2:** (a) Zonal mean meridional wind obtained by EXP1 (Lindzen's parameterization below 100 km height). Data are averaged from 1 June to 30 June. Negative and positive values are southward and northward wind, respectively. Contour intervals are 5 m s$^{-1}$. (b) As in Figure 2a except for EXP2 (Yigit's parameterization in the whole atmosphere). (c) Meridional wind difference between EXP1 and EXP2 (EXP2-EXP1). Contour intervals are 2 m/s.

**Figure 3:** (a) Zonal mean temperature obtained by EXP1 (Lindzen's parameterization below 100 km height). Data are averaged from 1 June to 30 June. Units are K. (b) As in Figure 3a except for EXP2 (Yigit's parameterization in the whole atmosphere). (c) Temperature difference between EXP1 and EXP2 (EXP2-EXP1). Contour intervals are 10 K.

**Figure4:** (a) Height-latitude distribution of the temperature component of the SW2 amplitude in June obtained by EXP1. Units are K. Contour intervals are 5 K. (b) As in Figure 4a except for EXP2. (c) Temperature difference between EXP1 and EXP2 (EXP2–EXP1).

**Figure 5:** (a) Height-latitude distribution of the zonal wind component of the SW2 amplitude in June obtained by EXP1. Units are m s$^{-1}$. (b) As in Figure 5a except for EXP2. (c) Zonal wind difference between EXP1 and EXP2 (EXP2–EXP1).

**Figure 6:** (a) Temperature component of the TW3 amplitude in June obtained by EXP1. Units are K. Contour intervals are 5K. (b)As in Figure 6a except for EXP2. (c) Temperature difference between EXP1 and EXP2 (EXP2–EXP1). Contour intervals are 2.5 K.

**Figure 7:** (a) Height-latitude section of the zonal mean of the zonal GW drag in June obtained by EXP1. Positive and negative values are eastward and westward acceleration, respectively. Units are m s$^{-1}$ day$^{-1}$. (b) As in Figure 7a except for EXP2.

**Figure 8:** (a) Height-longitude section of the zonal GW drag at 35° N obtained by EXP2. Data are averaged between 00 UT and 01 UT in June. Units are m s$^{-1}$ day$^{-1}$. (b) As in figure 8a except for zonal wind component. Units are m s$^{-1}$.

**Figure 9:** (a) Latitude-longitude section of the zonal GW drag at 120 km height in June. Data are averaged between 00 UT and 01 UT. Positive and negative values are eastward and westward acceleration, respectively. Units are m s$^{-1}$ day$^{-1}$. Contour intervals are 40 m s$^{-1}$ day$^{-1}$. (b) As in Figure 9a except for the average between 06 UT and 07 UT.

**Figure 10:** (a) Height-longitude section of the zonal GW drag at 65° N in June (EXP2). Data are averaged between 00 UT and 01 UT. Units are m s$^{-1}$ day$^{-1}$. Contour intervals are 50 m s$^{-1}$ day$^{-1}$. (b) As in figure 10a except for zonal wind component

obtained by EXP2. Units are m s$^{-1}$. (c) As in Figure 10b except for EXP1. (d) Difference of the zonal wind at 65° N between EXP1 and EXP2 (EXP2–EXP1). Units are m s$^{-1}$. Contour intervals are 10 m s$^{-1}$.

**Figure 11:** (a) Longitude–latitude section of the zonal GW drag at 200 km height in June. Data are averaged between 00 UT and 01 UT. Positive and negative values are eastward and westward acceleration, respectively. Units are m s$^{-1}$ day$^{-1}$. Contour intervals are 200 m s$^{-1}$ day$^{-1}$. (b) As in Figure 11a except for the average between 06 UT and 07 UT. (c) Vectors indicate the global distribution of the horizontal wind at 200 km height obtained by EXP2. Data are averaged between 00 UT and 01 UT in June. The vectors on the right-hand side indicate the zonal wind and meridional winds with magnitudes of 200 m s$^{-1}$. Color bars are the global distribution of the zonal wind component at 200 km height. Data are averaged between 00 UT and 01 UT. (d) As in Figure 11c except for the average between 06 UT and 07 UT.

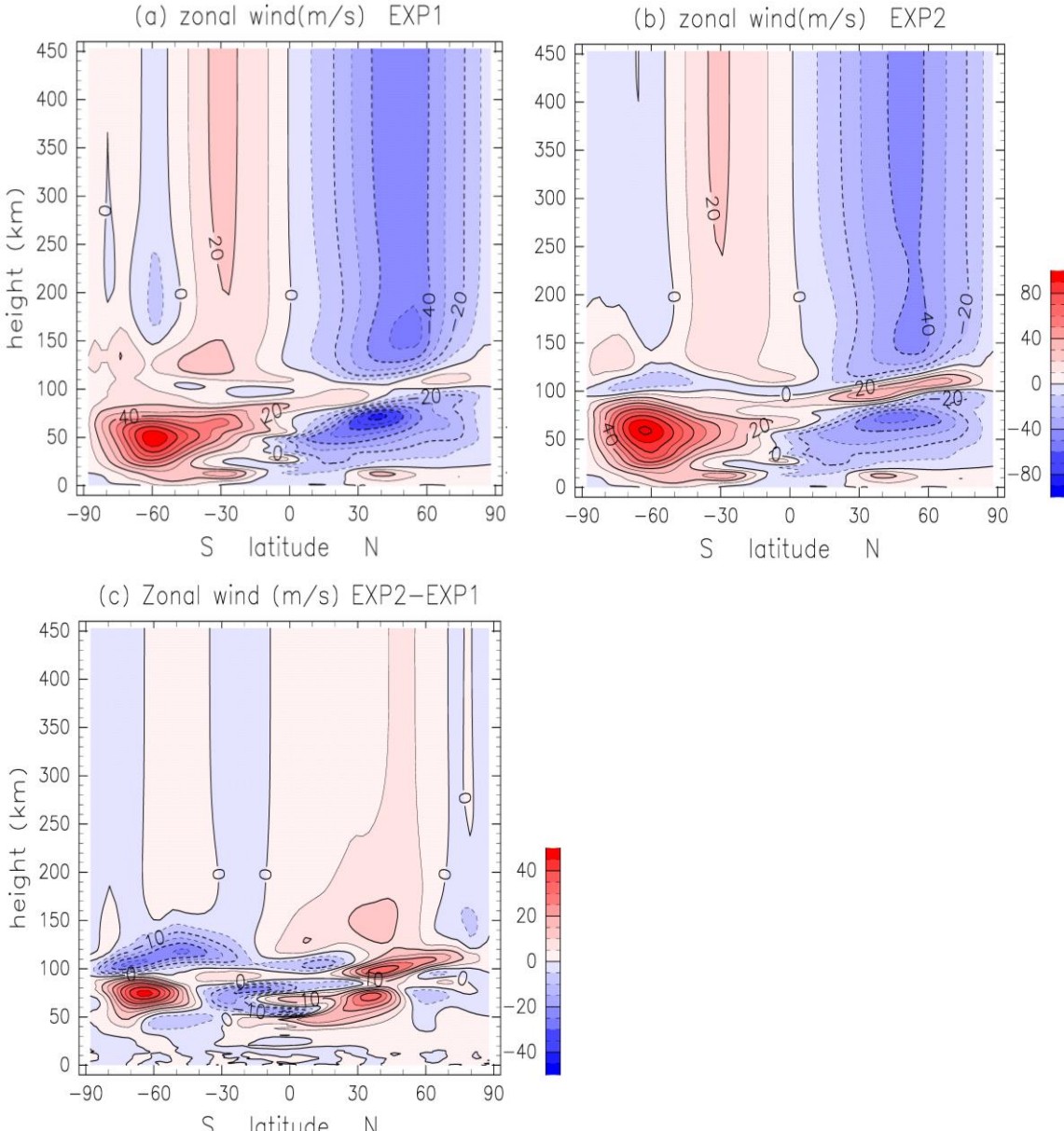

Figure1: (a) Height-latitude section of zonal and diurnal mean zonal wind obtained by EXP1 (the application of the Lindzen scheme below 100 km height). Data are averaged from 1 June to 30 June. Contour intervals of black lines are 10 m s⁻¹. Negative and positive values are eastward and westward winds, respectively. (b) As in Figure 1a except for the application of the Yiğit scheme in the whole

atmosphere (EXP2). (c) Difference of the zonal wind between EXP1 and EXP2 (EXP2–EXP1). Contour

intervals are 5 m/s.

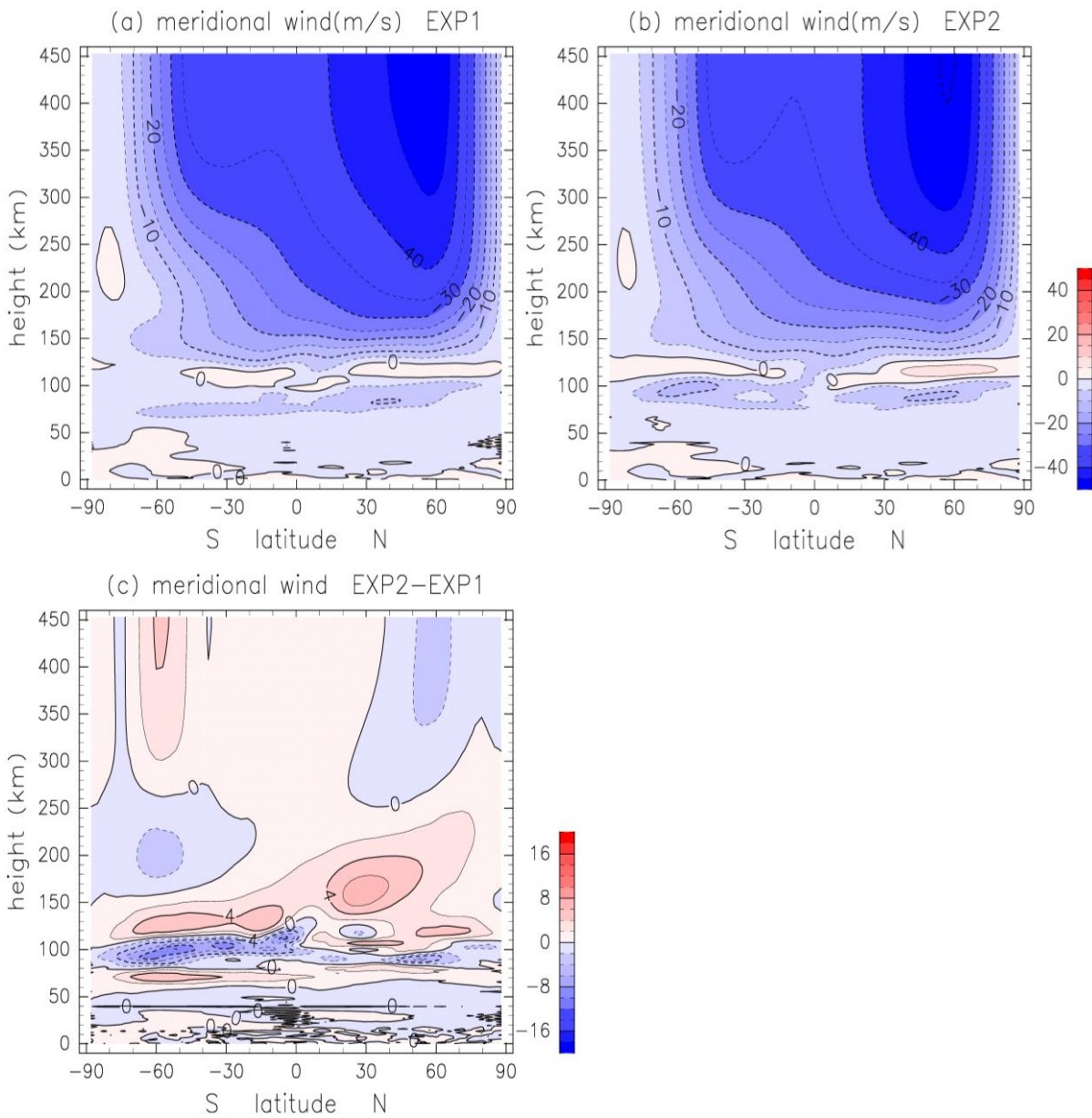

Figure2: (a) Zonal mean meridional wind obtained by EXP1 (Lindzen's parameterization below 100 km height). Data are averaged from 1 June to 30 June. Negative and positive values are southward and northward wind, respectively. Contour intervals are 5 m s$^{-1}$. (b) As in Figure 2a except for EXP2 (Yigit's parameterization in the whole atmosphere). (c) Meridional wind difference between EXP1 and

5   EXP2 (EXP2-EXP1). Contour intervals are 2 m/s.

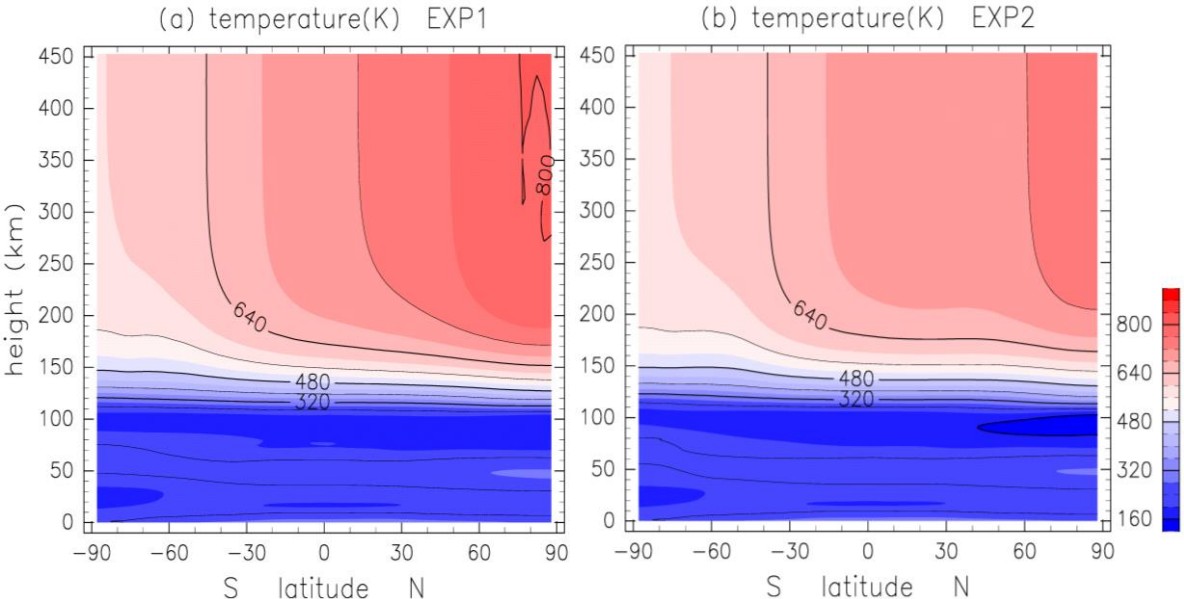

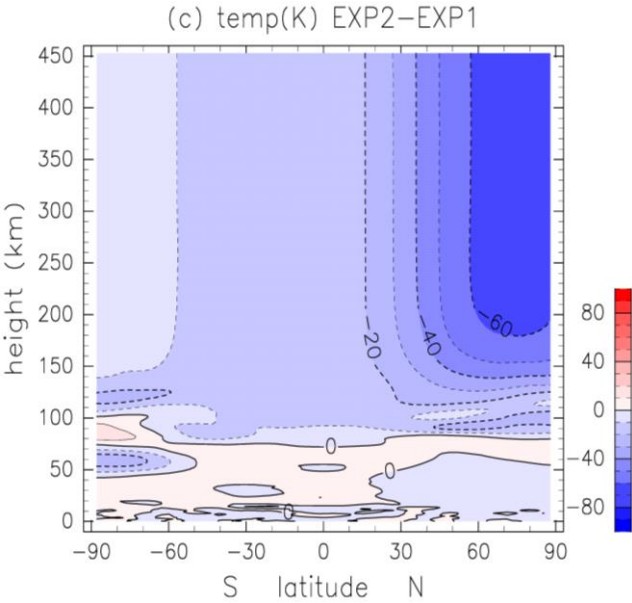

Figure 3: (a) Zonal mean temperature obtained by EXP1 (Lindzen's parameterization below 100 km height). Data are averaged from 1 June to 30 June. Units are K. (b) As in Figure 3a except for EXP2 (Yigit's parameterization in the whole atmosphere). (c) Temperature difference between EXP1 and EXP2 (EXP2-EXP1). Contour intervals are 10 K.

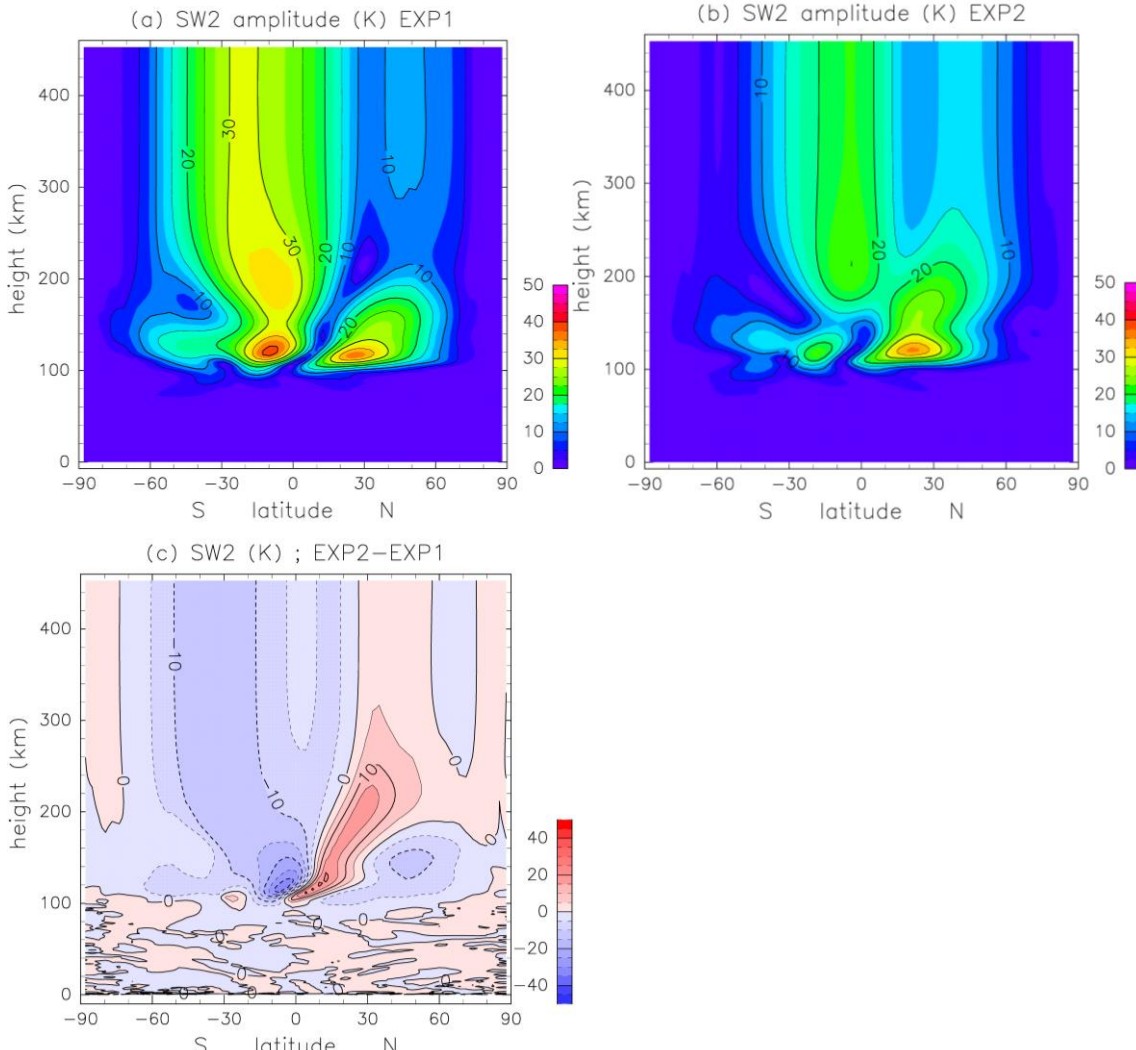

5 Figure4: (a) Height-latitude distribution of the temperature component of the SW2 amplitude in June
obtained by EXP1. Units are K. Contour intervals are 5 K. (b) As in Figure 4a except for EXP2. (c)
Temperature difference between EXP1 and EXP2 (EXP2–EXP1).

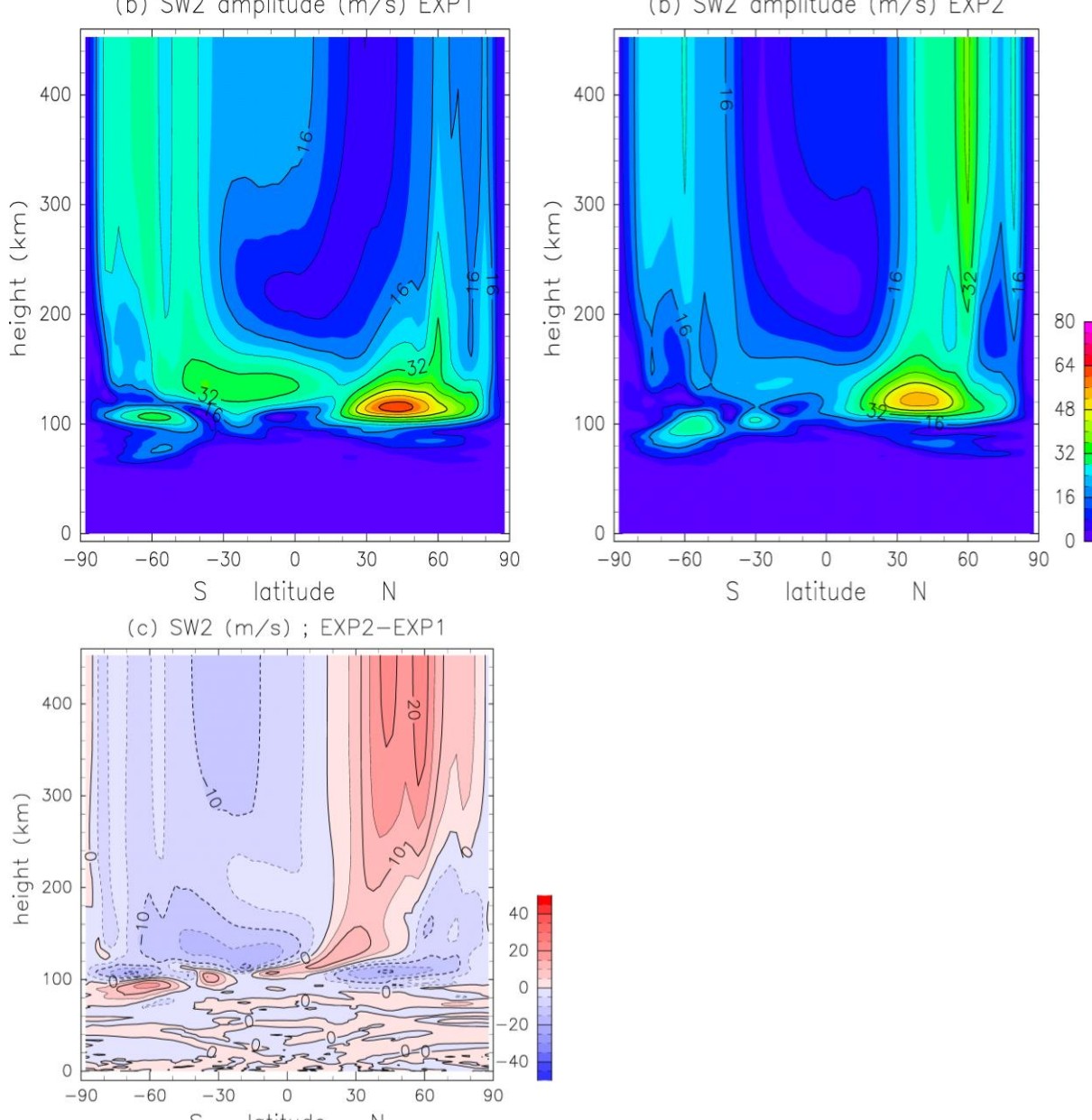

Figure 5: (a) Height-latitude distribution of the zonal wind component of the SW2 amplitude in June obtained by EXP1. Units are m s$^{-1}$. (b) As in Figure 5a except for EXP2. (c) Zonal wind difference between EXP1 and EXP2 (EXP2–EXP1).

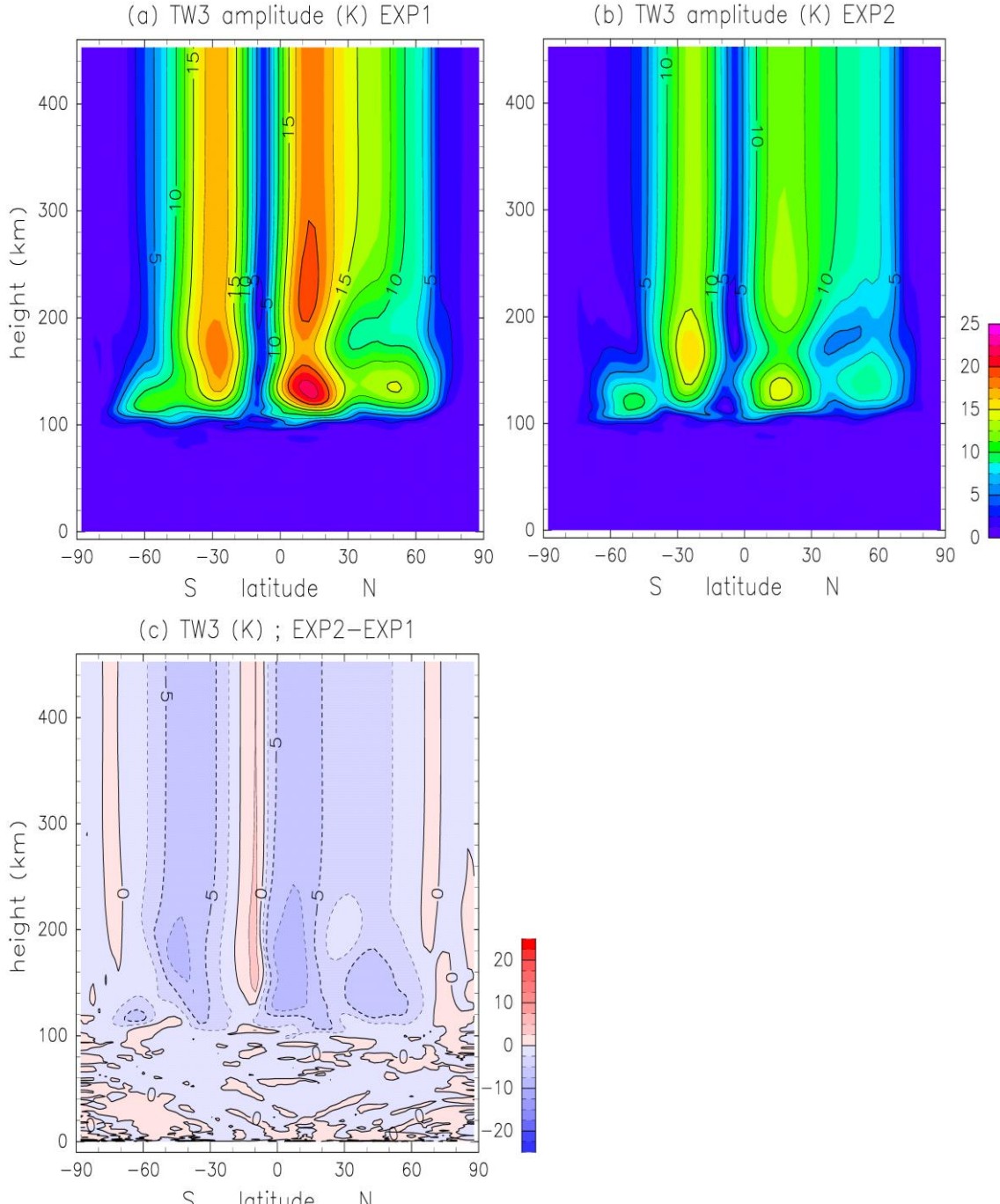

Figure 6: (a) Temperature component of the TW3 amplitude in June obtained by EXP1. Units are K. Contour intervals are 5K. (b)As in Figure 6a except for EXP2. (c) Temperature difference between EXP1 and EXP2 (EXP2–EXP1). Contour intervals are 2.5 K.

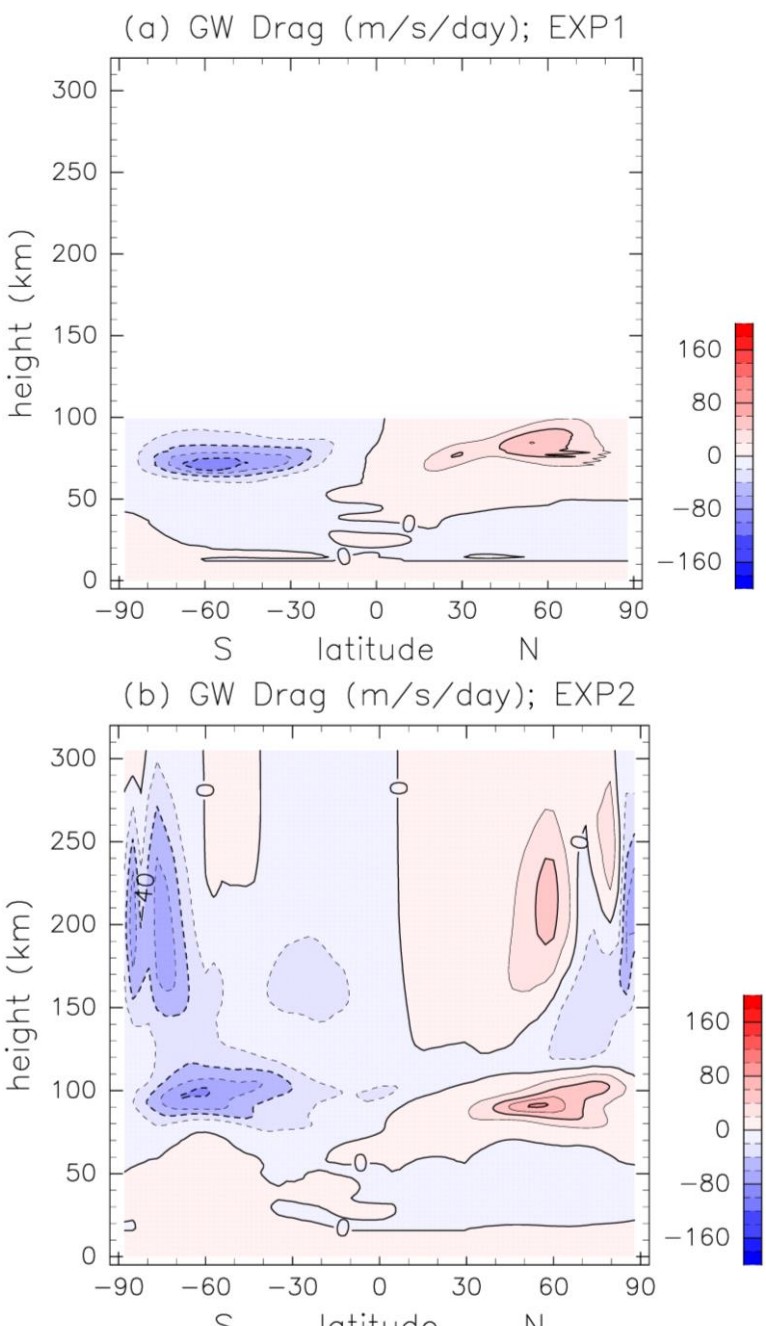

Figure 7: (a) Height-latitude section of the zonal mean of the zonal GW drag in June obtained by EXP1. Positive and negative values are eastward and westward acceleration, respectively. Units are m s$^{-1}$ day$^{-1}$. (b) As in Figure 7a except for EXP2.

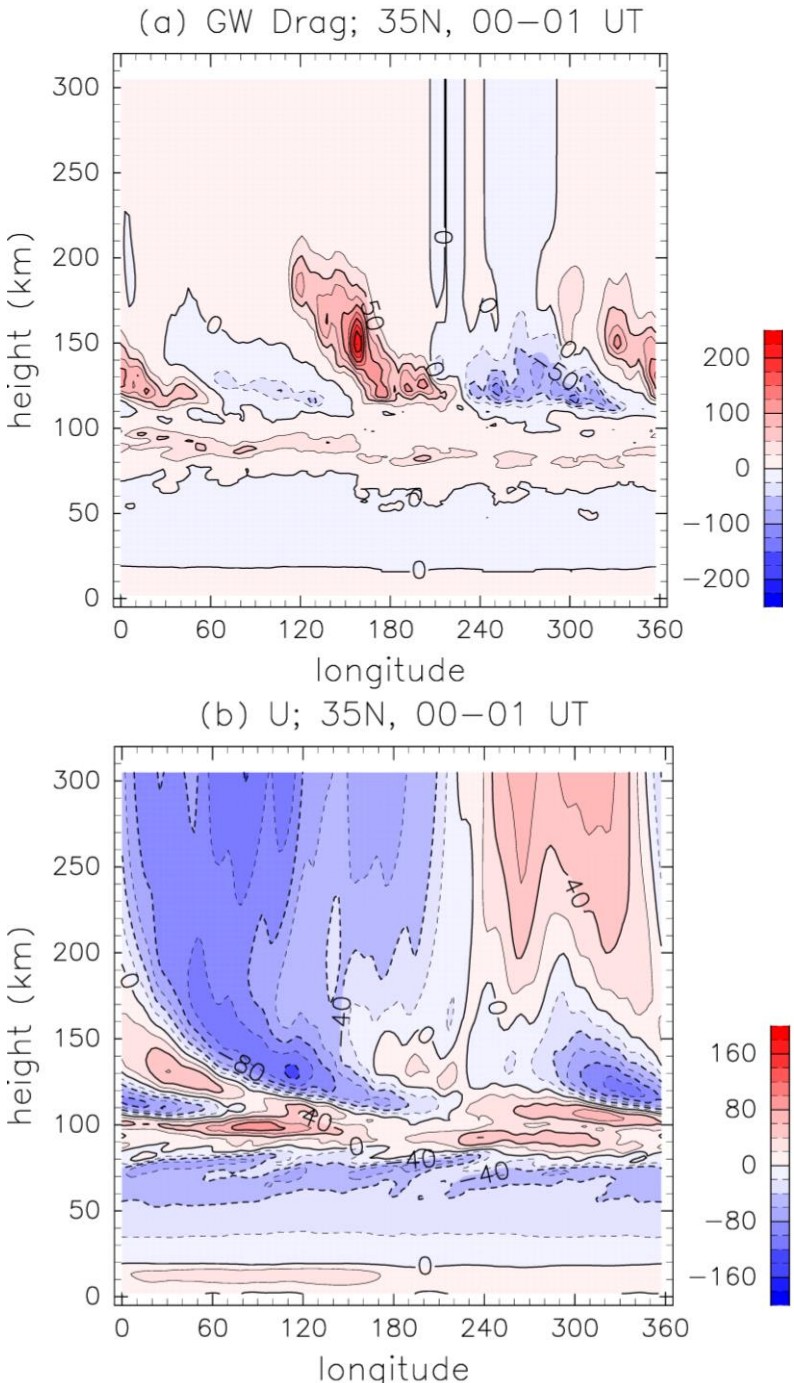

Figure 8: (a) Height-longitude section of the zonal GW drag at 35° N obtained by EXP2. Data are averaged between 00 UT and 01 UT in June. Units are m s$^{-1}$ day$^{-1}$. (b) As in figure 8a except for zonal wind component. Units are m s$^{-1}$.

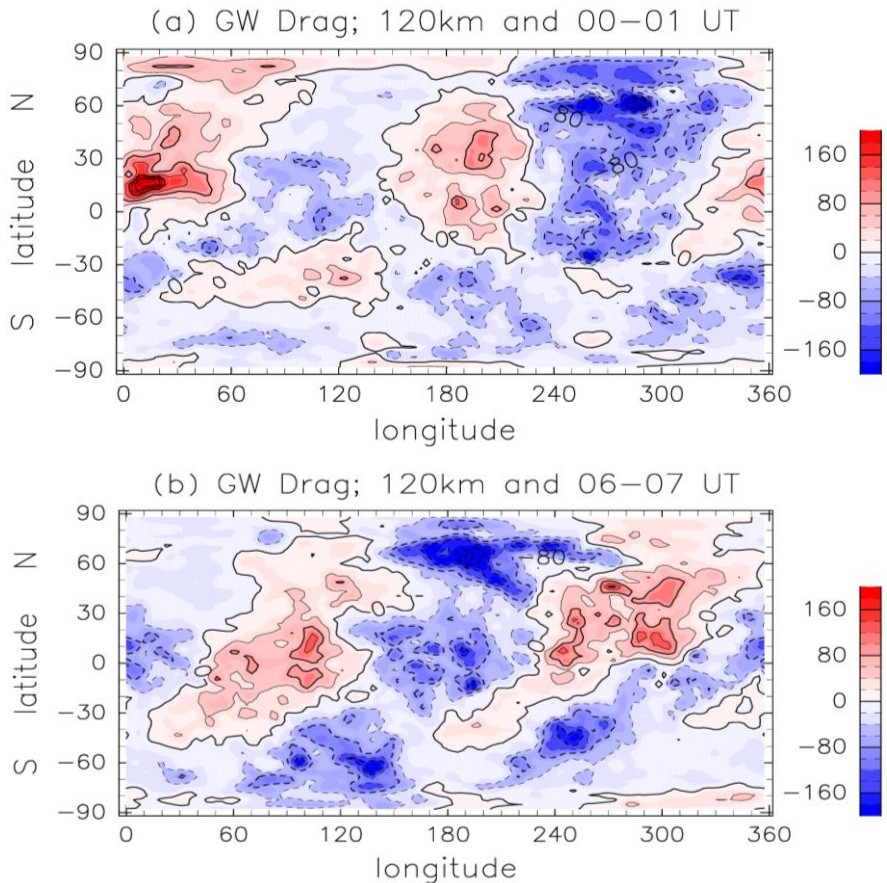

Figure 9: (a) Latitude-longitude section of the zonal GW drag at 120 km height in June. Data are averaged between 00 UT and 01 UT. Positive and negative values are eastward and westward acceleration, respectively. Units are m s$^{-1}$ day$^{-1}$. Contour intervals are 40 m s$^{-1}$ day$^{-1}$. (b) As in Figure 9a except for the average between 06 UT and 07 UT.

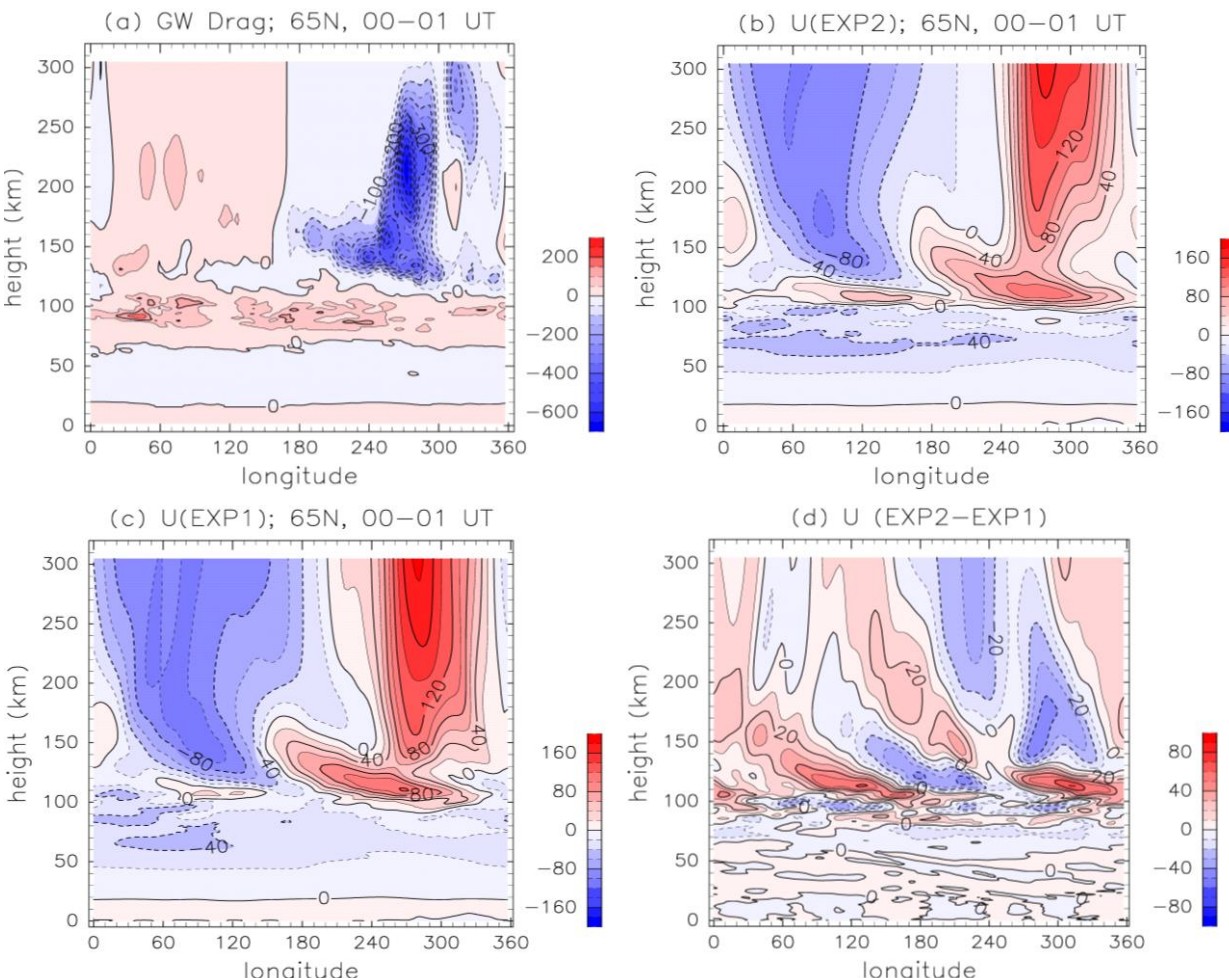

Figure 10: (a) Height-longitude section of the zonal GW drag at 65° N in June (EXP2). Data are averaged between 00 UT and 01 UT. Units are m s$^{-1}$ day$^{-1}$. Contour intervals are 50 m s$^{-1}$ day$^{-1}$. (b) As in figure 10a except for zonal wind component obtained by EXP2. Units are m s$^{-1}$. (c) As in Figure 10b except for EXP1. (d) Difference of the zonal wind at 65° N between EXP1 and EXP2 (EXP2–EXP1). Units are m s$^{-1}$. Contour intervals are 10 m s$^{-1}$.

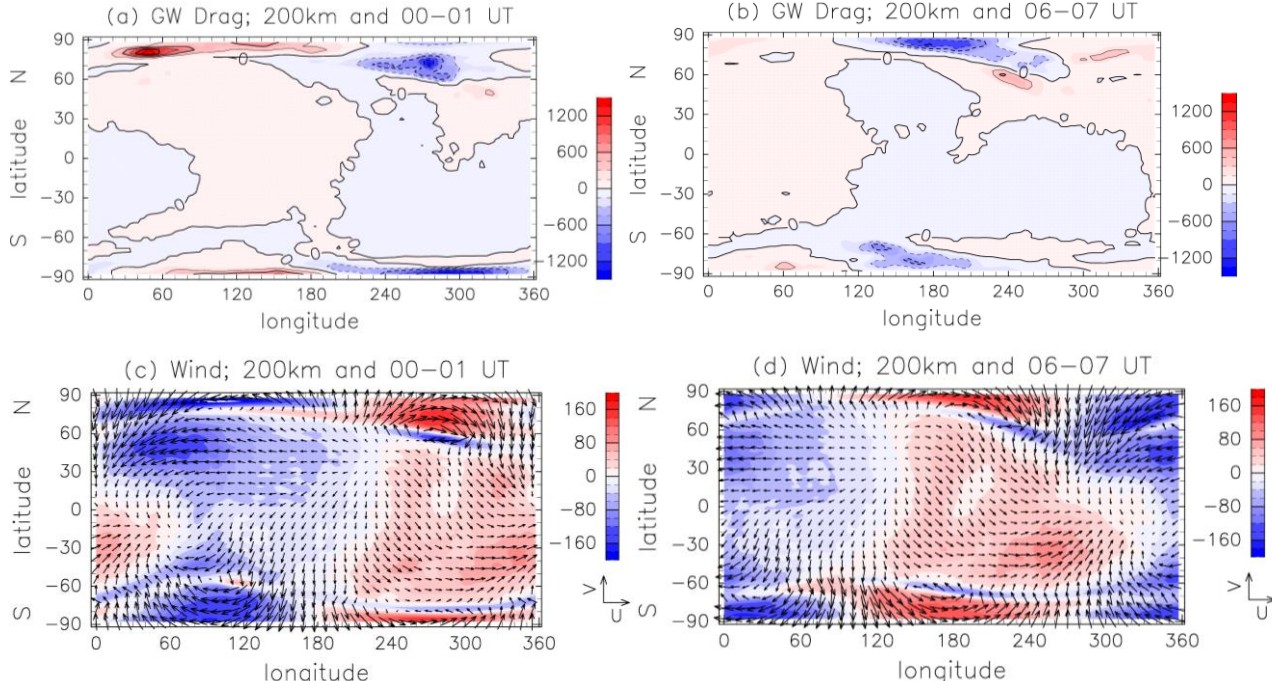

5 Figure 11: (a) Longitude–latitude section of the zonal GW drag at 200 km height in June. Data are averaged between 00 UT and 01 UT. Positive and negative values are eastward and westward acceleration, respectively. Units are m s$^{-1}$ day$^{-1}$. Contour intervals are 200 m s$^{-1}$ day$^{-1}$. (b) As in Figure 11a except for the average between 06 UT and 07 UT. (c) Vectors indicate the global distribution of the horizontal wind at 200 km height obtained by EXP2. Data are averaged between 00 UT and 01 UT in June. The vectors on the right-hand side indicate the zonal wind and meridional winds with magnitudes of 200 m s$^{-1}$. Color bars are the global distribution of the zonal wind component at 200 km height. Data are averaged between 00 UT and 01 UT. (d) As in Figure 11c except for the average between 06 UT and 07 UT.