# Peer review of "Impact of gravity wave drag on the thermospheric circulation: Implementation of a nonlinear gravity wave parameterization in a whole atmosphere model"

_Annales Geophysicae, 2019_

## Referee Comment (RC1) · Anonymous Referee #1 · 23 May 2019

Review of "Impact of gravity wave drag on the thermospheric circulation: Implementation of a nonlinear gravity wave parameterization in a whole atmosphere model" by Miyoshi and Yigit.

This manuscript describes initial results from a "whole atmosphere" general circulation model where an existing middle atmosphere gravity wave drag parameterization based on the Lindzen scheme is replaced with a parameterization based on "nonlinear interactions" following Medvedev and Klaassen that includes dissipation terms for thermospheric processes. The model simulations for perpetual June conditions

show that when a non-orographic GWD parameterization with thermospheric effects included (EXP2) is applied throughout the entire model domain, the resulting zonal mean zonal wind and amplitude of the migrating migrating tide (SW2) in temperature fields are quite different compared to results from similar model simulation using a Lindzen type GWD scheme from 0-100 km (EXP1). The authors claim that the SW2 in EXP2 is in better agreement with observations than EXP1 SW2 due to deceleration of the zonal wind by the thermospheric GWD. The authors conclude that parameterized GWD in the thermosphere plays an important role in the momentum budget of the thermosphere and is essential for low resolution whole atmosphere models to more realistically simulate the atmosphere-ionosphere system.

The manuscript presents interesting results from a whole atmosphere model highlighting the effects of parameterized GWD in the thermosphere on the zonal winds, which in turn affect the amplitudes of SW2. Neither the GCM or the GWD parameterization are new, but this seems to be the first time these tools have been used together to describe the effect of parameterized GWD on some aspects of the thermospheric circulation. The results of this investigation in whole atmosphere modeling would be of interest to Annales Geophysicae readership. However, I think in its present form the manuscript is incomplete regarding description of the methodology and discussion of the results. As a result, the authors do not reach substantial conclusions concerning the role of GWD in the thermosphere that can be supported with the results in the manuscript. I recommend major revisions, as described in detail below. Specifically, these revisions should: (1) provide more background information on GWD and whole atmosphere modeling; (2) include more detail regarding the experimental design and numerical methods used to analyze the model output; (3) describe the impact of the GWD parameterization on zonal mean temperature in the thermosphere in addition to zonal mean winds; (4) include the impact on the diurnal migrating tide (DW1); (5) describe in more detail the better agreement with observed SW2 from EXP2. I believe these revisions are necessary in order to provide a substantial contribution to this area of research.

Major comments:

(1) It would be helpful to the reader if the Introduction briefly described the approach of GWD parameterization and the challenges presented by extending these parameterizations from the middle atmosphere to the thermosphere, where different physical processes apply (e.g., page 4 lines 1-2). It may help to first cite a basic reference describing the difference between linear and nonlinear approaches (e.g., Fritts and Alexander, Rev. Geophysics, 2003 may be one such reference) to GWD parameterization and then cite and briefly describe in more detail why the thermospheric environment requires modified or addition physical terms. The introduction should also cite other related studies involving GCM simulations of the thermosphere at high resolution (e.g., Liu et al., GRL 2014 https://doi.org/10.1002/2014GL062468) or with parameterized GWD (Becker, JAS, 2017, https://doi.org/10.1175/JAS-D-16-0194.1, or England et al. JASTP 2006 https://doi.org/10.1016/j.jastp.2005.05.006), and describe how the present study fits in with previous work. These are just some examples, the authors no doubt are more aware of recent work in this field than I am, but the point is that there is already a fair amount of research in this area that should be noted in the manuscript.

(2) The experimental design needs more description. Specifically, a. what is the model time step? b. How long was the perpetual June simulation carried out? Is this just 30 days of simulation? Figure 1 caption states only 30 days of results were averaged (June 1-30). c. Is there any spin up to the model to reach a steady state? This is important to know; is the SW2 signal steady or varying strongly with time throughout the simulation? d. Can the authors explain why is there no experiment where the "linear" middle atmosphere scheme is applied above 100 km? It is not surprising that a simulation with parameterized GWD artificially cut off at 100km will produce a different thermospheric state than a simulation with parameterized GWD extending throughout the entire model domain. In its present form, the study isn't really telling us that the Yigit scheme is needed for a GCM in the thermosphere, it's just telling us that more drag on the winds gives a different SW2. This is rather superficial. To be more substantial,

it would help to know the benefit of the nonlinear Yigit scheme? What dissipation terms are important (nonlinear interaction, radiative damping, diffusion, ion drag, etc?). Running a simulation with the linear scheme in the thermosphere could answer this.

(3) Figure 1 plots zonal mean zonal wind, but not temperature. Since Figure 2 shows the SW2 in temperature, and since the authors claim the GW thermal effects are very important (page 2 line 6), it makes sense to add temperature to Fig 1 to give a complete description of the differences in the background thermospheric state between EXP1 and EXP2.

(4) Effects of GWD on SW2 are discussed, but not DW1 or other tidal modes, particularly nonmigrating tides. Why where these modes not considered? Earlier, Miyoshi et al (2014) used this exact same model at slightly higher resolution (1 degree latitude/longitude) and found that the diurnal tide is important above 200 km. Since these tides are not acting in isolation to one another, it makes sense to describe the impact of GW drag on both DW1 and SW2 at least. Please also mention in section 3.2 how the tidal amplitudes are obtained from the model.

(5) The description of the agreement between EXP2 SW2 and observations (Section 3.2, not 1.2 as in the manuscript) is not entirely convincing. SABER estimates are quoted as 15-20 K but it's not clear if this is for June conditions, over what years, or what kind of uncertainty is associated with this number. Does 15-20 K mean that is the typical range of values? Do the SW2 amplitudes from EXP1 and EXP2 vary widely over the simulation period? It would be most helpful if Figure 2 plotted SABER SW2 results as a function of latitude and altitude for June to provide a comprehensive comparison. The authors should also do the same for DW1 – that is, compare DW1 amplitude from EXP1, EXP2 and SABER.

---

## Referee Comment (RC2) · Anonymous Referee #2 · 12 Jun 2019

The paper describes an experiment of including a gravity wave parameterization routine for the whole atmosphere, i.e. including the thermosphere, in a general circulation model, thereby replacing the old, linear, parameterization scheme which is only active in the middle atmosphere. The authors show comparison of zonal mean winds, semidiurnal tidal parameters, and some figures of parameterized gravity wave drag and wind in the thermosphere. The authors conclude that the new routine provides a better zonal mean climatology, better representation of the semidiurnal tide. Having parameterized gravity wave in the thermosphere also makes it possible to investigate the interaction

between tides and gravity waves. The paper is generally well written, and the results may be of interest to the community. I recommend publication after some more minor modifications and additions.

Zonal means: The authors present only results for the zonal mean zonal wind. I would have liked to see temperature and meridional wind results as well. These are directly influenced by gravity waves, and play a crucial role e.g. for transports around the MLT.

Tides: the authors show semidiurnal tidal signatures. Are there any useful results for other tides, like diurnal or terdiurnal tide?

Minor issues

Title: Gravity → gravity

Abstract, L 17: dynamical → dynamical factor

Page 1, L 25: insert "the" before behaviour

Section 1.2, 1st paragraph: Forbes et al analyzed the SW2 in the exosphere, please describe that more clearly.

Page 3, model description: the gravity wave propagation strongly depends on the phase speed spectrum. Are there observational constraints for the selected spectrum?

Page 6, l 15/16: within the one-hour period. This indicates a change in time, better: at different longitudes

L18: shows A height-longitude

L24 THE Yigit

L24 mechanism → mechanisms

Page 7, L2: the diurnal variation is also significant at midlatitudes

L14/15: also in the mesosphere. Please describe in more detail

Page 8, L8/9 Recent studies. . . please provide a reference

Page 9, l 26: have → has

Page 10, reference Gavrilov et al.: nonlineareffects → nonlinear effects

Caption Fig. 5: . . .except for THE zonal. . .

---

## Author Comment (AC1) · 25 Jul 2019

Yasunobu Miyoshi and Erdal Yiğit

y.miyoshi.527@m.kyushu-u.ac.jp

Thank you for your constructive and helpful comments. According to your comments, we have revised manuscript.

(1)It would be helpful to the reader if the Introduction briefly described the approach of GWD parameterization and the challenges presented by extending these parameterizations from the middle atmosphere to the thermosphere, where different physical processes apply (e.g., page 4 lines 1-2). It may help to first cite a basic reference

describing the difference between linear and nonlinear approaches (e.g., Fritts and Alexander, Rev. Geophysics, 2003 may be one such reference) to GWD parameterization and then cite and briefly describe in more detail why the thermospheric environment requires modified or addition physical terms. The introduction should also cite other related studies involving GCM simulations of the thermosphere at high resolution (e.g., Liu et al., GRL 2014 https://doi.org/10.1002/2014GL062468) or with parameterized GWD (Becker, JAS, 2017, https://doi.org/10.1175/JAS-D-16-0194.1, or England et al. JASTP 2006 https://doi.org/10.1016/j.jastp.2005.05.006), and describe how the present study fits in with previous work. These are just some examples, the authors no doubt are more aware of recent work in this field than I am, but the point is that there is already a fair amount of research in this area that should be noted in the manuscript.

The introduction now cites some of the relevant references suggested by the reviewer. Also, we have discussed in the beginning of introduction GW parameterizations in GCMs in the context of the upper atmosphere physics above ∼105 km (turbopause). Overall, the rationale for the extension of GW parameterization to the thermosphere is discussed in a number of previous publications (e.g., YiÄğit et al., 2008; YiÄğit and Medvedev. (2013)

(2)The experimental design needs more description. Specifically, a. what is the model time step? b. How long was the perpetual June simulation carried out? Is this just 30 days of simulation? Figure 1 caption states only 30 days of results were averaged (June 1-30). c. Is there any spin up to the model to reach a steady state? This is important to know; is the SW2 signal steady or varying strongly with time throughout the simulation? d. Can the authors explain why is there no experiment where the "linear" middle atmosphere scheme is applied above 100 km? It is not surprising that a simulation with parameterized GWD artificially cut off at 100km will produce a different thermospheric state than a simulation with parameterized GWD extending throughout the entire model domain. In its present form, the study isn't really telling us

that the Yiħit scheme is needed for a GCM in the thermosphere, it's just telling us that more drag on the winds gives a different SW2. This is rather superficial. To be more substantial, it would help to know the benefit of the nonlinear Yiħit scheme? What dissipation terms are importantãĂĂ(nonlinear interaction, radiative damping, diffusion, ion drag, etc?). Running a simulation with the linear scheme in the thermosphere could answer this.

The descriptions of numerical simulation in the previous manuscript were incorrect. Numerical simulation started on 1 June, and we conducted 2-year numerical integration with seasonal variation. The GCM was nudged by Meteorological reanalysis data (JRA55) up to 40 km height to simulate realistic temporal variations in the lower atmosphere. The time step of the GCM is 30 s, and the data are sampled every 1 h during the numerical simulation. The data from 1 June to 30 June in the second year are analyzed in this study. The GCM has realistic temporal variations, so that the SW2 has significant day-to-day and seasonal variations.

The rationale for the extension of GW parameterizations into the thermosphere has been discussed in a number of research (Yiħit et al., 2008, 2009; Yiħit and Medvedev., 2013) and review papers (Yi?it and Medvedev, 2015; Yiħit et al., 2016). In fact, now the extended introduction summarizes the need for the whole atmosphere scheme, i.e., Yiħit et al. (2008) scheme, for GCMs extending in to the thermosphere. There are a number of differences between the conventional linear schemes and the nonlinear scheme of Yiħit et al. (2008). Linear schemes all use intermittency factors as they unrealistically overestimate GW drag, while the Yiħit scheme does not contain any intermittency or artificial factors. Second, while linear schemes assumes that waves propagates independently, as if other waves are not present, the Yiħit scheme accounts for the mutual interaction between different GW harmonics in the entire GW spectrum. An illustrative intercomparison of the linear approach with the nonlinear whole atmosphere approach of our GW scheme has already been performed in the initial work of Yiħit et al. (2008)..

(3)Figure 1 plots zonal mean zonal wind, but not temperature. Since Figure 2 shows the SW2 in temperature, and since the authors claim the GW thermal effects are very important (page2line6), it makes sense to add temperature to Fig1 to give a complete description of the differences in the background thermospheric state between EXP1 and EXP2.

The following sentences are inserted in the manuscript (section 3.1)

Figures 2a and 2b show the height–latitude distribution of the zonal mean meridional wind obtained by EXP1 and EXP2, respectively. In both experiments, southward flow from the summer pole to winter pole is dominant at 50–100 km height, whereas northward flow appears between 100 and 120 km height. These flows are stronger in EXP2 than in EXP1, which is explained by the enhanced GW drag in EXP2 as shown later. Above 130 km height, southward flow is dominant in both experiments. The magnitude of the southward wind between 130 and 250 km height is weaker in EXP2 than that in EXP1 except for southward of 30° S (Figure 3c). This weaker meridional wind in EXP2 is caused by the meridional component of the GW drag. On the other hand, the difference of meridional wind between EXP1 and EXP2 is small above 250 km height (less than 10 %). Figure 3a and 3b shows the height-latitude distribution of the zonal mean temperature obtained by EXP1 and EXP2, respectively. At 80–100 km height, cooling and warming occur at 30–90° N and at 60–90° S, respectively (Figure 3c). This cooling and warming effects are caused by the enhanced southward wind (meridional circulation) at 80–100km height in EXP2. Namely, the cooling (warming) at 30–90° N (60–90° S) is due to enhanced upward (downward) wind. It is noteworthy that cooling prevails above 100 km height. In particular, cooling at high latitudes in the NH exceeds 60 K. This cooling is caused by the GW thermal effect. This indicates that GW induced cooling also affects thermal structure in the upper thermosphere. Our results support the conclusion of the GCM work by YiÄğit and Medvedev (2012), who showed for the first time that GWs cool the thermosphere during low solar activity conditions.

(4)Effects of GWD on SW2 are discussed, but not DW1 or other tidal modes, particularly nonmigrating tides. Why where these modes not considered? Earlier, Miyoshi et al (2014) used this exact same model at slightly higher resolution (1 degree latitude/longitude) and found that the diurnal tide is important above 200 km. Since these tides are not acting in isolation to one another, it makes sense to describe the impact of GW drag on both DW1 and SW2 at least. Please also mention in section 3.2 how the tidal amplitudes are obtained from the model.

The impact of GWD parameterization on DW1 was studied in detail by in the work by Yiħit and Medvedev (2017). The impact on DW1 obtained in this study is similar to that presented by Yiħit and Medvedev. Therefore, we have not focused on DW1 tide in this study. Effects of GWD on TW3 are newly added. The following sentences are inserted in section 3.3.

Figure 6a shows the height–latitude distribution of the temperature component of the migrating terdiurnal tide (TW3) amplitude in June obtained by EXP1. The amplitude peak is located at $15°$ N latitude, and secondary peak appears at $25–30°$ S. The maxima are 23 K at $15°$ N and 130 km height, and 18 K at $17.5°$ S and 165 km height. Figure 6b shows the temperature component of the TW3 amplitude obtained by EXP2, and Figure 6c shows the amplitude difference between EXP1 and EXP2. The latitudinal structure of the TW3 in EXP2 is quite similar to that in EXP1. However, the amplitude is weaker in EXP2 than in EXP1 by about 20–40%. The amplitude difference is significant in the 120–220 km height range. The TW3 is also attenuated by the GWD. Forbes et al. (2008) indicated that the TW3 amplitude at 110 km height is between 5 and 8 K. However, there are only a few studies concerning the satellite observation of TW3 amplitude in the 120–220 km height range. A detailed comparison of the TW3 amplitude between the simulation and observation is a subject of a future study.

(5) The description of the agreement between EXP2 SW2 and observations (Section 3.2, not 1.2 as in the manuscript) is not entirely convincing. SABER estimates are quoted as 15-20 K but it's not clear if this is for June conditions, over what years, or what kind of uncertainty is associated with this number. Does 15-20 K mean that is the
typical range of values? Do the SW2 amplitudes from EXP1 and EXP2 vary widely over the simulation period? It would be most helpful if Figure 2 plotted SABER SW2 results as a function of latitude and altitude for June to provide a comprehensive comparison. The authors should also do the same for DW1 – that is, compareãĂĂDW1 amplitude from EXP1, EXP2 and SABER.

I think my description of the observed SW2 tide in the original manuscript is unclear. The following sentences are inserted in section 3.2.

Pancheva (2011, IAGA book) studied climatology (6-year mean from 2002 to 2007) of SW2 temperature tide using SABER observation. Figure 2.3 in Pancheva (2011) indicates that the SW2 in June at 110 km height has peaks at 20-30 N and 20 S. The maxima at 15-30 N and at 15-25 S are 25-28 K and 15-20 K, respectively. The peak values of the monthly mean SW2 amplitude in EXP2 at 110 km height are 26 K at 20N and 21 K at 15 S. The SW2 amplitude obtained in EXP2 is consistent with the SABER observation.

The SW2 also has significant day-to-day variations. For example, the SW2 amplitude at 20 N in EXP1 (EXP2) ranges from 27 (22) to 37 (31) K, and the standard deviation of day-to-day variations in the SW2 amplitude at 20 N in EXP1 and EXP2 are 2.8 K and 2.9 K, respectively. Similar day-to-day variations in the SW2 amplitude are found below 100 km height. This indicates that day-to-day variations in the SW2 amplitude are primarily generated in the lower atmosphere and propagates into the lower thermosphere.

Please also note the supplement to this comment:
https://www.ann-geophys-discuss.net/angeo-2019-36/angeo-2019-36-AC1-supplement.pdf

---

## Author Comment (AC2) · 25 Jul 2019

Thank you for your constructive and helpful comments. According to your comments, we have revised manuscript.

»Zonal means: The authors present only results for the zonal mean zonal wind. I would have liked to see temperature and meridional wind results as well. These are directly influenced by gravity waves, and play a crucial role e.g. for transports around the MLT.

We added effects of the GW drag on the zonal mean meridional wind and temperature

in section 3.1.

»Tides: the authors show semidiurnal tidal signatures. Are there any useful results for other tides, like diurnal or terdiurnal tide?

We added effects of the GW drag on the terdiurnal tide in section 3.3.

Minor issues >Title: Gravity > gravity Done >Abstract, L 17: dynamical ! dynamical factor Done >Page 1, L 25: insert "the" before behavior Done >Section 1.2, 1st paragraph: Forbes et al analyzed the SW2 in the exosphere, please describe that more clearly. ãĂĂ"in the upper thermosphere" was replaced by "in the exobase (400-500 km)".

>Page 3, model description: the gravity wave propagation strongly depends on the phase speed spectrum. Are there observational constraints for the selected spectrum?

The following sentence was added in the third paragraph of section 2. The GW spectrum adopted in this study and its relation to the observation have been discussed in detail in the work by YiÄ§it et al. (2008) and YiÄ§it et al. (2009).

>Page 6, l 15/16: within the one-hour period. This indicates a change in time, better: at different longitudes "one-hour" is typo. "one-hour" was replaced by "one-day".

>L18: shows A height-longitudeãĂĂDone >L24 THE Yigit Done >L24 mechanism > mechanisms Done >Page 7, L2: the diurnal variation is also significant at midlatitudes "significant" was replaced by "the largest".

>L14/15: also in the mesosphere. Please describe in more detail ãĂĂThe differences of the GW drag in the mesosphere were discussed in section 3.4.

>Page 8, L8/9 Recent studies: : : please provide a reference Papers by (Miyoshi and Fujiwara (2003, 2008) and Akmaev et al. (2008) are added.

>Page 9, l 26: have > has Done

[Figure]

>Page 10, reference Gavrilov et al.: nonlineareffects > nonlinear effects Done

>Caption Fig. 5: : : :except for THE zonal: : Done

Please also note the supplement to this comment:
https://www.ann-geophys-discuss.net/angeo-2019-36/angeo-2019-36-AC2-supplement.pdf